# Myasthenia Gravis and Thymectomy at a Tertiary-Care Surgical Centre: A 20-Year Retrospective Review

**DOI:** 10.3390/curroncol32120662

**Published:** 2025-11-27

**Authors:** Olivia Lauk, Alexandre Sarmento De Oliveira, Caroline Huynh, Sohat Sharma, Arthur Vieira, Michelle Mezei, Kristine Chapman, Hannah Briemberg, Kristin Jack, John Yee, Anna L. McGuire

**Affiliations:** 1Department of Surgery, Division of Thoracic Surgery, Vancouver General Hospital, Vancouver, BC V5Z 1M9, Canada; olivia.theisenlauk@vch.ca (O.L.); alexandre.sarmentodeoliveira@vch.ca (A.S.D.O.); sohat410@student.ubc.ca (S.S.);; 2Vancouver Coastal Health Research Institute, Vancouver, BC V5Z 1M9, Canada; 3Department of Surgery, Division of Thoracic Surgery, Centre Hospitalier Affilié Régional (CHAUR)—Trois-Rivières, Université de Montréal—Campus Trois-Rivières, Trois-Rivières, QC G8Z 3R9, Canada; 4Department of Medicine, Division of Neurology, Vancouver General Hospital, Vancouver, BC V5Z 1M9, Canada; 5BC Cancer Research, Vancouver, BC V5Z 4E6, Canada; 6Department of Surgery, University of British Columbia, Gordon & Leslie Diamond Health Care Centre, 2775 Laurel Street #7109, Vancouver, BC V5Z 1M9, Canada

**Keywords:** myasthenia gravis, thymoma, mediastinal mass, thymectomy, remission

## Abstract

Myasthenia gravis (MG) is an autoimmune disease that often affects patients with abnormalities of the thymus gland, including thymomas. Surgical removal of the thymus (thymectomy) is an important treatment for both thymomatous and non-thymomatous MG, yet questions remain regarding which patients benefit most. In this 20-year retrospective study from a large tertiary-care center, we analyzed 420 thymectomies to examine the relationship between thymoma subtype, MG occurrence, and remission after surgery. We found that younger age, thymic hyperplasia, and WHO type B2 thymoma were strongly associated with MG. Importantly, remission rates six months after thymectomy were similar between patients with thymomatous and non-thymomatous MG. Minimally invasive video-assisted thoracoscopic surgery (VATS) resulted in shorter hospital stays and fewer complications than open surgery, although, it must be emphasized, that open surgery was performed in the majority of the cases. These findings support thymectomy as an effective treatment option for MG and highlight the importance of individualized surgical strategies based on patient and tumor characteristics.

## 1. Introduction

The most common anterior masses are known as the 4T’s: thymoma, teratoma, thyroid tissue, and ‘terrible’ lymphoma [1]. Thymic epithelial tumors, such as thymoma and thymic carcinomas, are responsible for up to 1% of all malignancies. They present the vast majority of all anterior mediastinal masses [1,2,3]. Myasthenia gravis (MG), a rare autoimmune antibody-mediated disorder, is commonly associated with thymoma in 21%, according to a meta-analysis of 49 studies by Mao and colleagues, whilst the reported incidence rates in the current literature show a broad variety [4]. MG shows altered neuromuscular transmission due to the targeting of acetylcholine receptor (AChR) and muscle receptor-associated proteins, such as muscle-specific tyrosine kinase (MuSK) and low-density lipoprotein receptor-related protein 4 (LRP4) [5]. It is characterized by fluctuating muscle weakness and fatiguability involving variable combinations of ocular, bulbar, limb, and respiratory muscles. Therapies include acetylcholinesterase inhibition (for symptom control), immunosuppression, immunomodulation, and ultimately thymectomy [5,6,7,8].

It is assumed that thymoma negatively influences the patient’s prognosis and plays an important role in the pathophysiology of MG [4]. The recommendations/guidelines regarding the optimal treatment approach for patients with epithelial thymic tumors, are based on series, trials of mainly open surgical approach. Therefore, randomized controlled trials answering the question of the optimal surgical approach are currently nonexistent [9,10].

We hypothesize that patients with thymomatous MG exhibit distinct histopathological characteristics and are not significantly different in terms of remission rates at 6 months post-thymectomy compared to patients with non-thymomatous MG.

The aim of this study is to describe a large tertiary center with 20 years of experience of thymectomy in thymomatous and non-thymomatous MG patients, including the incidence of thymomatous MG, surgical approach, histopathological features in patients with and without MG, and additionally, remission/recurrence rates in patients with thymomatous vs. non-thymomatous MG patients.

## 2. Materials and Methods

### 2.1. Study Design

This is a cross-sectional retrospective cohort study of all consecutive video-assisted thoracoscopic surgery (VATS) or open trans-thoracic thymectomies performed for any reason at Vancouver General Hospital between 1 January 2001 and 31 December 2021. Adult male and female patients presenting for elective thymectomy were included. Patients without medical records indicating that a thymectomy was performed were excluded.

### 2.2. Data Sources

Patients, numbering 422, were identified through the Vancouver General Hospital 3M discharge abstract database by searching for the ICD-9 and ICD-10 codes “thymoma,” “thymic cancer,” “thymic mass,” “mediastinal mass,” and “myasthenia gravis,” under the diagnosis option (Figure 1, Flowchart). An anonymous password-protected Excel spreadsheet of study participants was produced, with an independent consecutive study number assigned to each participant record. To increase data accuracy and quality, all demographic, clinical, pathological, recurrence, and remission data were supplemented by individual patient chart review. A subsequent retrospective review of the Division of Neurology’s electronic medical record system was also queried to collect MG medical management data and follow-up. The following data points were collected: demographics (age, sex), pre-operative tumor characteristics (size, biopsy), pre-operative treatment, operative treatment (surgical approach and procedure), pathological characteristics (size, histology, WHO classification, and Masaoka–Koga stage for thymoma), post-operative data (length of stay, complications, treatment, remission, and recurrence). Information on ethnicity was not routinely collected, thus not available for analysis. This project was approved by the University of British Columbia’s (UBC) Clinical Research Ethics Board (REB), # H22-00426.

### 2.3. Objectives and Outcomes

The primary objective of this study was to determine the incidence of thymomatous MG for patients undergoing elective thymectomy and to highlight the differences of video-assisted thoracoscopic surgery (VATS) and open approach. Secondary objectives were to describe thymic background tissue histopathological characteristics for patients with and without MG, as well as remission/recurrence rates in patients with thymomatous and non-thymomatous MG.

Complications were classified according to the Clavien–Dindo classification system [11]. To evaluate the association between the presence of thymoma and MG, as well as to compare clinical characteristics between MG patients with and without thymoma, the chi-square test and Fisher’s exact test were used for categorical variables. Remission of MG was defined as the absence of MG symptoms with no need for continued MG-specific medication (e.g., pyridostigmine, corticosteroids, or immunosuppressants) following thymectomy.

Recurrence other than MG was defined as the reappearance of the tumor on imaging studies. Mortality was defined as death from any cause at any time during the follow-up period.

### 2.4. Statistical Analysis

Based on previous numbers of thymectomies conducted at our tertiary-care center and the nature of this rare entity of myasthenia gravis and thymic tumors, we expected a range of 20 to 30 thymic resections to be conducted annually. Therefore, for a 20-year retrospective review, we estimated between 400 to 600 cases to be available in the medical records for review.

Descriptive statistics were used to assess the study population. Univariate and multivariate regression statistics were used for tumor histopathological features, classification, and stage and were stratified according to presence and absence of MG. For patients whose pathology reports listed two thymoma WHO subtypes, the subtype with the higher grade was used for coding in this analysis. Multivariate regression analyses were performed to account for confounding factors when looking at characteristics associated with thymomatous MG, or all MG. l. Categorical variables were compared using Fisher’s exact test or the chi-square test, as appropriate. A *p*-value < 0.05 was used to determine statistical significance.

Statistical analyses were completed using SPSS 30 software.

## 3. Results

A total of 420 thymectomies (Table 1, Figure 2) were conducted in a population with an average age of 54.4 years (SD 16.1), and 59% were female. The thymic mass average mean size on a pre-operative computed tomography scan (CT scan) was 52.2 mm (SD 58.9 mm). A total of 56.2% were thymomas, more specifically WHO AB (32.6%) (Appendix A) and Masaoka–Koga stage I (39.8%), or IIa (44.1%) (Appendix A), and thymic cysts (14.3%). MG involved 39.5% of cases, with an incidence of 48.8% thymomatous MG (Table 1). To provide greater context, we present a detailed breakdown of thymic epithelial tumors (TETs) histology, stratified by MG status, in Appendix A.

In the multivariate regression analysis (Table 2 and Table 3), increasing age (OR 0.97, 95% CI 0.95–0.99, *p* < 0.001) and larger tumor size on pathology (OR 0.98, 95% CI 0.96–0.99, *p* < 0.001) were significantly associated with reduced odds of MG. Pre-operative biopsy was also negatively associated with MG (OR 0.3, 95% CI 0.13–0.68, *p* = 0.004). Longer hospital length of stay was associated with higher odds of MG (OR 1.07, 95% CI 1.01–1.12, *p* = 0.019). Regarding histology, thymoma subtype B2 (OR 2.92, 95% CI 1.22–6.98, *p* = **0.016**) and germinal hyperplasia (OR 5.5, 95% CI 2.47–12.27, *p* < 0.001) showed significantly higher odds of MG, while other histologic subtypes did not reach statistical significance (Table 3).

When comparing thymomatous and non-thymomatous MG patients, there was no statistically significant difference in 6-month complete remission rates post-thymectomy (67.7% vs. 67.1%, respectively; χ^2^(1) = 0.005, *p* = 0.941) (Appendix A).

## 4. Discussion

In this, to our knowledge, largest cross-sectional retrospective study, we investigated the incidence of MG associated thymomas/non-thymomas and their optimal treatment strategy. In our cohort, the incidence of thymomatous MG among all thymectomy patients was 19.2%, and among MG patients, 48.8% had thymomatous MG. These findings are in line with previous reports showing wide variability in the association between thymoma and MG [9,10,12]. For example, Kondo et al. reported that MG occurred in 24% of patients with thymoma, while others, such as Sanders et al., noted thymic pathology—including thymomas and hyperplasia—in up to 15% of MG patients. While not directly comparable due to differences in study design and population, our data support the significant overlap between thymoma and MG. [13].

In general, thymectomy is recommended for patients with thymic tumors (thymoma, thymic carcinoma, thymic neuroendocrine tumor) and for certain patients with MG symptoms. For thymomatous MG specifically, thymectomy has been considered the main modality of treatment regardless of whether MG is generalized, bulbar, or ocular [14].

In 2016, the international multicenter, rater-blinded randomized trial of thymectomy in myasthenia gravis, MGTX, confirmed improved clinical outcomes for patients undergoing thymectomy + prednisone with myasthenia gravis compared to sole prednisone therapy [15]. Of these 126 randomized patients results showed lower myasthenia gravis scores over a 3-year period, lower requirement of medication, and lower hospitalization rates for MG exacerbations in the thymectomy + prednisone group [15]. In 2019, the same group published a follow-up study showing continued benefits for these patients at 5 years, including good functional score and low requirements of medication [16]. Since then, total thymectomy with medical management for even non-thymomatous MG has been recommended by the updated International Consensus Guidance for Management of Myasthenia Gravis [14]. Of 166 MG patients in this cohort with and without underlying thymoma, 130 patients were still on MG medications with at least prednisone or mestinon.

A few retrospective studies [17,18] presented favorable results for minimal invasive surgeries compared to open surgery due to a shorter hospital stay, as well as lower morbidity and mortality rates as known for VATS procedures in general regardless of the underlying cause for surgery [17,19]. Most of our cohort underwent an open approach which is attributable to the long observation period of 20 years and the majority were performed before standardized minimal invasive approach was established at our institution. The median length of stay was 5 days in the overall population considering that more patients underwent open surgery as well as having a Masaoka stage of IIA/IIB—being 52% of 236 patients with thymic epithelial tumor pathology (Appendix A). Taking only the patients into account since the inauguration of our minimal invasive program, the median length of stay is reduced to 3 days. To lower the post-operative complication rate, complete resection should be targeted over partial resection as explored in several studies [18,20]. Our institutional overall post-operative complication rate was 44.8% and 21.2% (open surgery vs VATS), according to the Clavien–Dindo classification (Appendix A). The intraoperative complication rate for the minimal invasive approach was 1.5% (Appendix A), which is lower than the reported with up to 6% in the literature, compared to 11.5% intraoperative complications undergoing open surgery [21,22]. 

As also confirmed by our multivariate analysis, specific tumor and patient characteristics are independently associated with the presence of MG and thymomatous MG. Our findings indicate that younger age and smaller tumor size are independently associated with the presence of MG, consistent with previous observations that MG often presents earlier in the disease course. Importantly, histologic subtype B2 was significantly associated with MG, reinforcing the known link between cortical thymoma components and autoimmunity, as already described by Marx and colleagues [23]. These results underscore the need for precise histological characterization and may guide both pre-operative assessment and post-operative surveillance strategies in MG patients.

According to the latest guidelines, updated in 2020, thymectomy is recommended in patients with generalized AChR-positive MG, if they fail to respond to medical management, even in the absence of any thymomatous mass [14,24]. In patients with generalized AChR-negative MG, thymectomy may also be considered with the exception for patients with MuSK or LPR4 receptors [14,24]. In addition, there have been discussions of thymectomy in selected seropositive and seronegative MG patients with isolated severe ocular myasthenia gravis refractory to medical management [14].

Among MG patients with available serological data (n = 107), 74.8% were either AChR or MuSK antibody-positive. Of the patients with thymoma and serology (n = 52), 36.4% were AChR positive. No patient in the cohort tested positive for LRP4 antibodies (Appendix A).

Additionally, we elaborated that thymoma, specifically WHO B2 thymoma (*p* = 0.016), were more often found with MG. This coincides with the fact that multiple pathological studies have found that thymic germinal centers contain a widely heterogenous population of B-cells undergoing clonic proliferation, somatic hypermutation, and selection [25]. Furthermore, the incidence of germinal centers correlates with anti-AchR titer as well [26]. Additionally, thymic follicular hyperplasia is the result of proliferation of lymphatic vessels and high endothelial venules. Therefore, the presence of such physiological occurrences indicates active uptake of lymphocytes from the blood stream [26].

This study has limitations including its retrospective nature and the potential for inconsistent or incomplete data collection over the 20-year span. Additionally, the predominance of open surgical approaches in earlier years reflects institutional practice patterns and may not reflect current minimally invasive standards. However, the use of uni- and multivariate analyses in the statistical methods helped to minimize confounding factors. Future studies accounting for this bias are recommended to establish the true benefit of thymectomy in this patient population.

## 5. Conclusions

This is one of the largest studies identifying MG in 39.5% of resected thymomas over a 20-year period. It also suggests that WHO Type B2 thymoma were more likely to be associated with MG and reported similar remission rates for thymomatous vs. non-thymomatous MG at 6 months post-thymectomy. Given the strong association between specific histological features—particularly WHO B2 thymoma and germinal hyperplasia—and MG, future research may focus on exploring underlying immune mechanisms, including T-lymphocyte involvement, in the pathophysiology of thymomatous MG as well as new biological therapies in addition to surgery [27].

## Figures and Tables

**Figure 1 curroncol-32-00662-f001:**
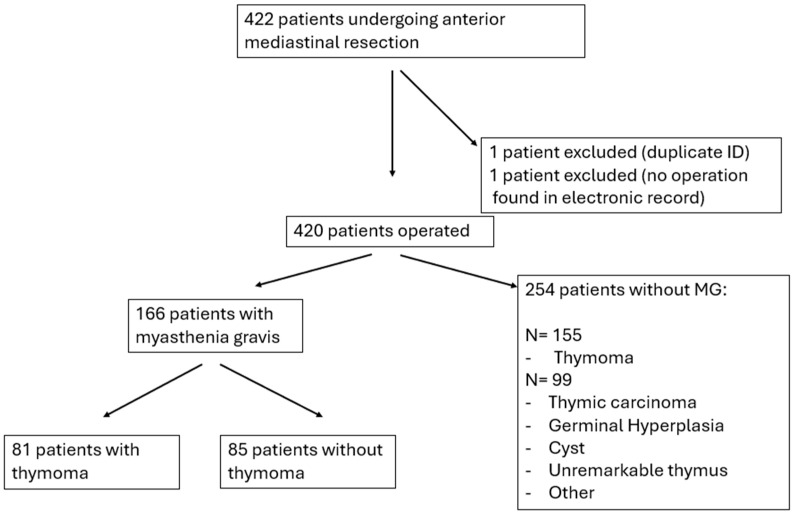
Flowchart.

**Figure 2 curroncol-32-00662-f002:**
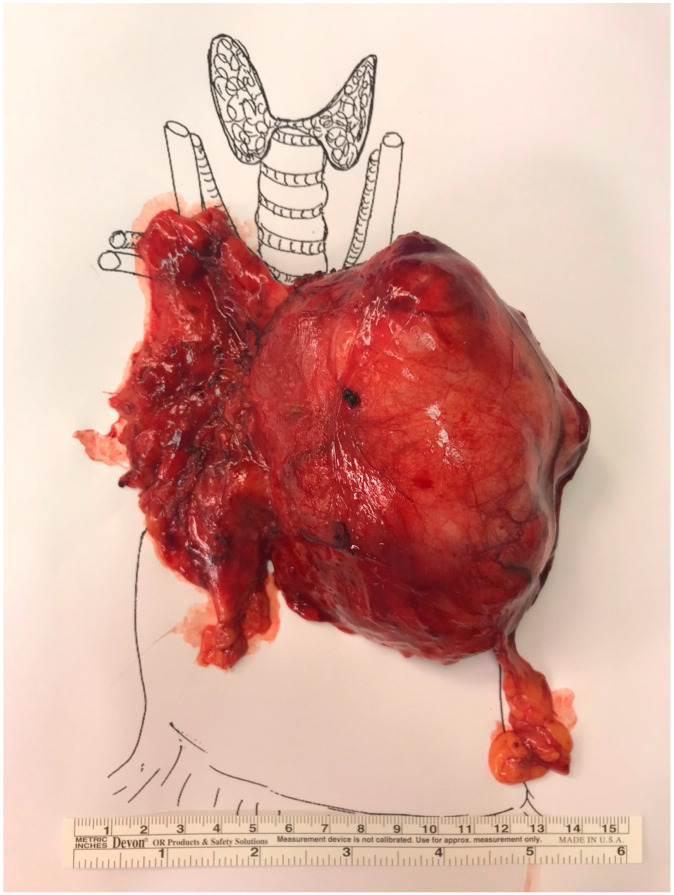
Total thymectomy and thymoma specimen oriented as per the International Thymic Malignancy Interest Group (ITMIG) surgical standards.

**Table 1 curroncol-32-00662-t001:** Patient and tumor characteristics stratified according to presence or absence of myasthenia gravis (MG), from 2001–2021.

Variable	Overall (*n* = 420)	MG (*n* = 166)	Non-MG (*n* = 254)	*p*-Value
**Female Sex (%)**	248 (59)	99 (59.6)	149 (58.6)	0. 842
**Age (yrs, mean ± SD)**	54.4 ± 16.1	49.2 ± 16.7	57.8 ± 14.7	**0.001**
**Pre-operative biopsy—*n* (%)**	70 (16.7)	11 (6.6)	59 (23.2)	**0.001**
**Post-operative complications—*n* (%)**	156 (37.1)	58 (35.2)	98 (38.4)	0.497
**30-day mortality** **Missing—*n* (%)**	30 (7.1) 33 (7.9)	9 (5.5) 14 (8.5)	21 (8.5) 19 (7.5)	0.292
**CT Tumor size (mm, mean ± SD)** **Missing—*n* (%)**	52.3 ± 58.8 107 (25.4)	42.5 ± 22.4 76 (46.1)	56.1 ± 67.7 30 (11.8)	**0.008**
**Length of Stay (days, mean ± SD)**	6.11 ± 5.76	6.89 ± 6.17	5.60 ± 5.43	**0.001**
**Thymoma Histology (%)**	236 (56.2)	81 (48.8)	155 (61)	0.016
**Thymic Carcinoma**	11 (2.6)	2 (1.2)	9 (3.5)	0.213
**Hyperplasia**	63 (15)	45 (27.1)	18 (7.1)	**0.001**
**Thymic Cyst**	60 (14.3)	11 (6.6)	49 (19.2)	**0.001**

Legend: Myasthenia gravis (MG); Age in years. CT = computed tomography. A *p*-value of <0.05 was statistically significant.

**Table 2 curroncol-32-00662-t002:** Multivariate logistic regression analysis of variables associated with myasthenia gravis. Variables significant in univariate analyses, including the presence of thymoma and clinical factors, were included to calculate adjusted odds ratios controlling for confounders.

Variable	OR	95% CI for OR	*p*-Value
Age (per year)	0.97	0.95–0.99	0.001
Sex (female vs. male)	1.22	0.70–2.10	0.484
Pathological tumor size (per mm)	0.98	0.96–0.99	<0.001
Pre-operative biopsy	0.30	0.13–0.68	0.004
Thymoma (vs. non-thymoma)	5.53	2.58–11.85	<0.001
Thymic carcinoma (vs. non-thymoma)	2.43	0.43–13.86	0.317
Length of stay (per day)	1.07	1.01–1.12	0.019
Post-operative complications	1.07	0.61–1.89	0.807
Surgical approach (VATS vs. open)	0.45	0.21–0.98	0.044

Legend: OR = odds ratio. CI = confidence interval. VATS = video assisted thoracoscopic surgery. A *p*-value of <0.05 was statistically significant.

**Table 3 curroncol-32-00662-t003:** Multivariate logistic regression analysis of clinical and histologic factors associated with myasthenia gravis.

Variable	OR	95% CI for OR	*p*-Value
Age (per year)	0.98	0.96–0.99	**0.024**
Sex (female vs. male)	1.11	0.68–1.80	0.675
Thymic carcinoma	0.62	0.12–3.39	0.585
A thymoma	1.13	0.38–3.41	0.823
AB thymoma	0.46	0.18–1.15	0.095
B1 thymoma	2.13	0.94–4.81	0.069
B2 thymoma	2.92	1.22–6.98	**0.016**
B3 thymoma	1.82	0.77–4.30	0.171
Hyperplasia	5.50	2.47–12.27	**<0.001**
Cyst	0.46	0.19–1.12	0.086
Calcification	4.41	0.92–21.18	0.064

Legend: OR = odds ratio. A *p*-value of <0.05 was statistically significant.

## Data Availability

The data supporting the findings of this study are available from the corresponding author upon reasonable request. Due to patient privacy and ethical restrictions, individual-level data are not publicly available.

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
