# Peer review of "Myasthenia Gravis and Thymectomy at a Tertiary-Care Surgical Centre: A 20-Year Retrospective Review"

_curroncol, 2025, doi:10.3390/curroncol32120662_

Round 1
Reviewer 1 Report
Comments and Suggestions for Authors
The authors performed a retrospective analysis of 420 patients who underwent thymectomy. Their aim was to clarify the relationship between thymoma subtype and MG incidence, as well as post thymectomy remission outcomes. They found that type B2 thymoma and germinal hyperplasia were more commonly associated with MG and that 6-month complete remission rates did not differ between thymomatous and non-thymomatous MG. Although it is a retrospective study from single institute, it is still precious data because of the large number of patients. I think it is well written, however, I would like the authors to revise in some minor points.
・The last line of page 2: The authors described that they use “Masaoka stage for thymoma”, however, there are some modified versions of Masaoka stage. I believe they used “Masaoka-Koga stage”. Please confirm it.
・The last line of “Result”: The authors wrote, “When comparing thymomatous and non-thymomatous MG, there was no difference in complete MG remission rate 6-month post-thymectomy, however, where did they show the data?
・First paragraph of “Discussion”: The authors wrote, “Our cohort confirmed these findings”, however, I did not feel so. In this paragraph, they showed their data of incidence of thymomatous MG in all of their cohort was 18%. According to the authors, Sanders, Comacchio, and Aljaafari reported that the incidence of thymoma or thymic hyperplasia in MG patients is 15%. Kondo reported that the incidence of MG in thymoma cases is 24%. These are very different and difficult to compare. Moreover, they also reported that the incidence of thymoma in MG patients was 48.8% in their cohort. It is not close to the data of Sanders or other authors.
・Page 7: The authors wrote, “In our patient cohort only 74% of our MG patient cohort were MuSK or AChR positive and of those 64% were associated with thymoma. Interestingly, none in the overall population were LPR positive”. Where did they show these data?
・”Conclusion” section: The authors wrote, “Given these results, future research will help clarify the role of T-lymphocyte immune response in the pathophysiology of thymomatous MG”. Is this conclusion possible to be led from their data?
Reviewer 2 Report
Comments and Suggestions for Authors
I have read and reviewed the article “Myasthenia Gravis and Thymectomy at a Tertiary-Care Surgical Centre: a 20-Year Retrospective Review”. I would like to congratulate the authors on this retrospective study that is well written. There are a few areas that require more data and discussion, specifically the exact number of patients with the various WHO classification types of thymomas would greatly help the manuscript.
Points to Address:
- Table 2 might be more impactful and provide context if there was a new and separate table of only patients with TETs (thymoma and thymic carcinoma) to report exactly how many of each and under “Thymoma” a detailed table of the WHO classification. This can be separated by MG and non-MG.
- The authors note that the LOS (length of stay) changed significantly between open and VATS approach. As this study was done over 20 years it is highly likely that the “case mix” changed over the two decades going from nearly all open in 2001 to a majority being done VATS by 2021. Therefore, if LOS is going to be evaluated in a multivariate analysis should the authors consider a time dependent analysis such as a sensitivity analysis using a year cut off when the case mix tipped to something like when at least half were done VATS? (or something similar)
- Recent manuscripts extoll the value of thymomyomectomy in patients with non-MG thymomas (PMID: 35723542 and 29268390) as a viable treatment option. Did the authors ever perform this in the series? If not, this should probably be at least mentioned in the discussion.
Points to Consider:
- There seems to be a formatting issue at lines 122-123. Please review.
